# Mediterranean Diet in Older Adults: Cardiovascular Outcomes and Mortality from Observational and Interventional Studies—A Systematic Review and Meta-Analysis

**DOI:** 10.3390/nu16223947

**Published:** 2024-11-19

**Authors:** Michela Furbatto, Diana Lelli, Raffaele Antonelli Incalzi, Claudio Pedone

**Affiliations:** 1Research Unit of Geriatrics, Department of Medicine and Surgery, Università Campus Bio-Medico di Roma, 00128 Rome, Italyc.pedone@policlinicocampus.it (C.P.); 2Operative Research Unit of Geriatrics, Fondazione Policlinico Universitario Campus Bio-Medico, Via Alvaro del Portillo 200, 00128 Rome, Italy; 3Operative Research Unit of Internal Medicine, Fondazione Policlinico Universitario Campus Bio-Medico, Via Alvaro del Portillo 200, 00128 Rome, Italy; r.antonelli@policlinicocampus.it; 4Research Unit of Internal Medicine, Department of Medicine and Surgery, Università Campus Bio-Medico di Roma, Via Alvaro del Portillo 21, 00128 Rome, Italy

**Keywords:** mediterranean diet, mortality, cardiovascular disease

## Abstract

Background/Objectives: there is conflicting evidence on the role of the Mediterranean Diet (MD) in reducing the risk of long-term outcomes in older adults. The aim of our study was to assess the effectiveness of high adherence to MD in reducing all-cause mortality and cardiovascular outcomes among older adults. Methods: PubMed database was searched up to 31 May 2023. We included randomized controlled trials (RCT) and cohort studies in the English language which evaluated the Mediterranean diet’s adherence to exposure on a population with a mean age > 60 years. The main outcomes were cardiovascular fatal and non-fatal events, and all-cause mortality. A sub-analysis on individuals > 70 years old was conducted. Meta-analysis of Observational Studies in Epidemiology (MOOSE) and Preferred Reporting Items for Systematic reviews and Meta-Analyses (PRISMA) guidelines were used for assessing data quality and validity. Pooled data were obtained by using random-effects models. Results: a total of 28 studies were included in this meta-analysis (26 observational studies and 2 randomized trials), reporting a total of 679,259 participants from different continents. Our results showed that high adherence to the MD reduces all-cause mortality risk by 23% (95% CI: 0.70–0.83), while it decreases the risk of cardiovascular mortality by 27% (95% CI: 0.64–0.84) and that of non-fatal cardiovascular events by 23% (95% CI: 0.55–1.01). Conclusions: MD is a promising dietary pattern for promoting health among older adults, as it is associated with reduced risks of all-cause and cardiovascular mortality, and non-fatal cardiovascular events. Adopting a Mediterranean Diet may contribute to better overall health and a lower likelihood of cardiovascular-related health issues in older individuals.

## 1. Introduction

Cardiovascular mortality is a leading cause of mortality and disability among adults [1] and the promotion of healthy nutrition is an established prevention strategy [2,3,4]. The Mediterranean Diet (MD) [5,6], is associated in the general population with a decreased risk of all-cause and cardiovascular mortality, but also of cardiovascular non-fatal events [3,4,7,8]. Definitive data on MD’s impact on older adults’ global health are conflicting, due to the fact that studies focusing on older adults have small sample sizes [9,10,11,12], as they generally include middle-aged adults and older patients representing a minority of the study population [12,13].

Dietary interventions at an older age may not be as effective as the data may suggest. Physiological changes occur during the aging process [12,13] combined with the loss of appetite, cognitive decline, and chewing problems which render nutritional interventions challenging in older demographics [14].

Moreover, aging is an independent risk factor for cardiovascular disease [15], but other risk factors are compounded by additional factors including frailty [16], obesity [17] and diabetes [18] which are common conditions in an older population. These risk factors are known to complicate and enhance cardiac risk factors in an older demographic [19], therefore the positive effect of MD observed in the general population may not be directly extrapolated to an older demographic.

The aim of the present study is to undertake a systematic review and a meta-analysis of the literature to assess the effectiveness of high adherence to MD in reducing cardiovascular events, cardiovascular mortality, and all-cause mortality among older adults.

## 2. Materials and Methods

This systematic review and meta-analysis were performed and reported according to the Preferred Reporting Items for Systematic Reviews and Meta-Analyses (PRISMA) statement.

PubMed database research was conducted by two independent researchers (M.F., D.L.), to identify eligible studies that were published from 1 January 1995 to 31 May 2023.

The search strategy consisted in using combinations of the following terms: “Mediterranean diet” and “cardiovascular”, “mortality” and “Mediterranean diet”, “Mediterranean diet” and “elderly”, “Mediterranean diet” and “aging”, “mediterranean diet” and “frailty”, “Mediterranean diet” and “survival”.

The inclusion criteria were: (1) MD adherence as exposure; (2) mean age of the study population >60 years of age or sub-analysis for participants over the age of 60; (3) all-cause mortality, cardiovascular mortality, and/or cardiovascular non-fatal events as study outcome; (4) randomized controlled trials and cohort studies as study design; (5) English language.

Any disagreements involved in the study selection were determined through a consensus meeting between the two researchers and, if necessary, a third senior researcher (C.P.) resolved discrepancies.

The study outcomes were all-cause mortality, cardiovascular mortality, and overall cardiovascular events.

Systematic information extraction encompassed publication details, study design, population characteristics, and outcome-related data. Hazard ratios (HR), or risk ratios (RR) with respective 95% confidence intervals (95% CI) were extracted from each study regarding study outcomes. Some of the included studies [20,21,22,23,24] had multiple intervention groups for each outcome due to the incorporation of nuts or olive oil into a standard Mediterranean Diet. Each of the intervention groups has been included and analyzed individually in the present study.

In accordance with the MOOSE guidelines, The Newcastle–Ottawa Quality Assessment Scale (NOS) [25] was used to assess the quality of the observational studies included. Additionally, following the PRISMA guidelines, the Cochrane Risk of Bias 2 (RoB 2) tool [26] was employed to evaluate the quality of the randomized controlled trials (RCTs).

A detailed protocol was prepared and registered in INPLASY (INPLASY2024100072, DOI number 10.37766/inplasy2024.10.0072).

For all the study outcomes, we calculated the standard error of each study. Pooled RR with 95% CI using random-effects models was then obtained and graphically represented using forest plots. For studies with multiple treatment arms, we used a three-level approach as suggested by Cheung that takes into account the nesting of comparisons within the same study [27]. Studies for which standard error was not calculable were excluded from the meta-analysis.

A subgroup analysis for subjects over the age of 70 was conducted, in order to allow us to expand our current knowledge to this segment of population which is more susceptible to cardiovascular events compared to younger persons.

It was not possible to perform a sensitivity analysis including only randomized controlled trials (RCT) due to the small number of these studies. Heterogeneity across studies was evaluated using the *I*^2^ statistics. *I*^2^ > 50% was considered to indicate high heterogeneity. Publication bias was assessed by inspecting the symmetry of funnel plots.

All statistical analyses were carried out using R software version 4.1.2 (R Foundation for Statistical Computing, Vienna, Austria).

## 3. Results

The process of study search and study selection is shown in Figure 1.

A total of 3808 publications were identified through PubMed database search. After title and abstract screening, 145 were assessed for eligibility. A total of 117 were excluded after full-text review because (1) the mean age of the study population was under 60 years; (2) the exposure did not include MD; (3) the outcomes did not include all-cause mortality or cardiovascular events (fatal and non-fatal); (4) the article was unavailable in the English language; (5) data were partially incomplete; (6) the article was a duplicate; (7) other reviews and/or a meta-analysis. Ultimately, a total of 28 studies were included in the meta-analysis, which included 26 observational studies and 2 randomized trials, reporting a total of 679,259 participants from different countries and continents. No studies were excluded for unobtainable standard error.

### 3.1. Adherence to MD and All-Cause Mortality

Characteristics of the selected individual studies are presented in Table 1.

A total of 19 studies evaluated all-cause mortality as primary outcome; most studies were based in European countries, while there were 3 in the United States [40,41,43] and 1 in Australia [39] with a mean follow up time of 104 months, ranging from 12 to 168 months. A total of 158,520 subjects were involved, 39.4% of which were male with a mean age of 69 years (ranging from a mean age of 60 to 82.9 years old). All selected studies applied an ordinal scale to assess the adherence to MD, and we compared high vs. low adherence to MD. Potential confounders commonly taken into account were sex, age, body mass index, education, total energy intake, smoking status.

The overall risk of bias analysis demonstrated a moderate-to-low risk, indicating a moderate-to-high quality of the included studies (Appendix A).

The RRs of each study and the pooled risk estimate are shown in Figure 2.

A higher adherence to MD was associated with a 23% decrease in the risk of all-cause mortality (pooled RR: 0.77; 95% CI: 0.70–0.83). Heterogeneity across studies was high (*I*^2^: 82%; *p*-heterogeneity < 0.01). Funnel plots visual analysis confirmed the presence of heterogeneity among the studies, as well as a potential publication bias (Appendix A).

The sub-analysis on participants above the age of 70, included 9 studies [9,29,30,31,32,35,40,42,44] and 10,827 participants, 55% of which were male, with a mean age of 75.3 years. The included studies were mostly (*n*-8) based in European countries, while 1 was based in the United States of America and 1 in Australia with mean follow up time of 89.7 months.

In this subpopulation there was a 18% decrease in the all-cause mortality risk for higher MD adherence (pooled RR: 0.82; 95% CI: 0.76–0.88), with a high heterogeneity between studies (*I*^2^: 60%; *p*-heterogeneity: 0.01) (Appendix A).

Visual assessment of funnel plots identified asymmetry, which confirms high heterogeneity among the studies, as well as a potential publication bias as shown in Appendix A.

### 3.2. Adherence to MD and Overall Cardiovascular Events

Characteristics of the selected individual studies are presented in Table 1.

Fifteen studies analyzed cardiovascular diseases as an outcome. Most studies (*n*-8) were based in European countries, (*n*-5) in the United States [41,43,47,48,51] and (*n*-2) in Hong Kong [50,52]. A total of 14 were cohort studies and 2 were randomized controlled trials with a mean follow-up time of 94 months, ranging from 7.7 to 168 months. A total of 568,911 participants were included, 45.5% of which were male; the mean age was 67 years (ranging from 60 to 80 years old).

All selected studies applied an ordinal scale to assess the adherence to MD. Potential confounders most frequently taken into account were sex, age, body mass index, education status, total energy intake, smoking status.

The overall risk of bias analysis demonstrated a moderate-to-low risk, indicating a moderate-to-high quality of the included studies (Appendix A).

The RRs of each study and the pooled risk estimate are shown in Figure 3.

A high adherence to MD was associated with a 25% decrease in the risk of cardiovascular events (pooled RR: 0.75; 95% CI: 0.68–0.83). Heterogeneity across studies was high (*I*^2^: 67%; *p*-heterogeneity: <0.01). Funnel plots visual analysis confirmed the presence of heterogeneity among the included studies, as well as a potential publication bias (Appendix A).

The sub-analysis on subjects above 70 years of age included 5 studies [9,29,30,45,52] and a total of 10,504 participants, 55.1% of which were male, with mean age 73.8 years old. The included studies were mostly (*n*-4) based in European countries, while one in Hong Kong had a mean follow-up time of 80.2 months.

A similar risk reduction was documented in this population, despite it not reaching statistical significance (pooled RR: 0.73; 95% CI: 0.49–1.06). Heterogeneity across studies was high (*I*^2^: 72%; *p*-heterogeneity < 0.01) (Appendix A).

Visual assessment of funnel plots highlighted asymmetry, which confirmed high heterogeneity among the included studies (Appendix A).

### 3.3. Adherence to MD and Cardiovascular Mortality

The characteristics of the selected individual studies are presented in Table 1.

A total of nine studies analyzed cardiovascular fatal events as an outcome. Most studies were based in European countries [23,37,46,49] and three in the United States [43,47,48]. Most (*n*-5) had a cohort design and *n*-1 was randomized controlled trial with a mean follow-up time of 108 months, ranging from 79 to 125 months. A total of 151,170 participants were included, 18% of which were male; the mean age was 64.8 years (ranging from 60 years old to 69 years old).

All selected studies applied an ordinal scale to assess the adherence to MD. Potential confounders more frequently taken into account were sex, age, body mass index, education status, total energy intake, and smoking status.

The overall risk of bias analysis demonstrated a moderate-to-low risk, indicating a moderate-to-high quality of the included studies (Appendix A).

The results of the association between adherence to the MD and cardiovascular mortality are shown in Figure 4.

A high MD adherence was associated with a 27% decrease in the risk of cardiovascular mortality (pooled RR: 0.73; 95% CI: 0.64–0.84). Heterogeneity across studies was high (*I*^2^: 64%; *p*-heterogeneity: <0.01). Funnel plots visual analysis confirmed the presence of heterogeneity among the included studies (Appendix A).

We were unable to perform a sub-analysis for individuals >70 years of age for this specific outcome because only one study [29] satisfied our eligibility criteria.

### 3.4. Adherence to MD and Non-Fatal Cardiovascular Events

The characteristics of the selected individual studies are presented in Table 1.

Seven studies analyzed cardiovascular non-fatal events as an outcome. Four studies were based in European countries [9,30,37,45], one in the United States [48], two in Hong Kong [50,52]. Most (*n*-4) had cohort design and one was a randomized controlled trial with a mean follow-up time of 50 months, ranging from 7.7 to 108 months. A total of 13,264 participants were included, 52.1% of which were male; the mean age was 69.8 years (ranging from 67 years old to 73 years old).

All selected studies applied an ordinal scale to assess the adherence to MD. Potential confounders commonly taken into account by adjusted models were sex, age, body mass index, education status, total energy intake, and smoking status.

The overall risk of bias analysis demonstrated a moderate-to-low risk, indicating a moderate-to-high quality of the included studies (Appendix A).

The results of the association between adherence to the MD and cardiovascular mortality are shown in Figure 5.

High adherence to MD was associated with a 23% decrease in cardiovascular non-fatal events (pooled RR: 0.75; 95% CI: 0.55–1.01). Heterogeneity across studies was high (*I*^2^ = 67%; *p*-heterogeneity < 0.01). Funnel plots visual analysis confirmed the presence of heterogeneity among the included studies (Appendix A).

The sub-analysis on participants aged > 70 years old included four studies [9,30,45,52], with a total of 9580 participants, 50.7% of which were male with mean age of 74.5 years.

Most studies (*n*-3) were based in European countries, while one was based in Hong Kong. The mean follow-up time was 80.2 months. Similar risk reductions were found, however, none were statistically significant (pooled RR: 0.72; 95% CI: 0.43–1.21). Heterogeneity across the studies was confirmed high (*I*^2^ = 82%; p-heterogeneity < 0.01) (Appendix A). The visual assessment of funnel plots highlighted asymmetry and confirmed high heterogeneity among the included studies, as shown in Appendix A.

## 4. Discussion

The present study showed that a higher adherence to MD is associated with a reduction in the risk of all-cause mortality, cardiovascular mortality, and cardiovascular non-fatal events in a population of older adults. In subjects above 70 years of age, we could not perform a meta-analysis on fatal cardiovascular events; with respect to the other outcome of interest, the risk reduction was statistically significant only for all-cause mortality.

These results confirm previous findings indicating a reduced risk of all-cause mortality among individuals with a high adherence to MD [51,53] and extends them to older people. In fact, a previous meta-analysis of 29 cohort studies [11] based on a healthy general population aged 18 years or older, showed that each 2-point increment in adherence to a MD was associated with 10% reduction in the risk of all-cause mortality. The association remained significant after stratification for gender, study location, follow-up duration, sample size, MD definition, and dietary assessment. However, the overall quality of the evidence was rated as low due to high inter-study heterogeneity, therefore lowering confidence in the effect estimates. Moreover, results were based on a general adult population and not focused primarily on older patients.

We observed a significant association between adherence to MD and cardiovascular fatal and non-fatal events in older adults, albeit not statistically significant in subjects above 70 years of age. Our findings extend to a geriatric sample of previous data from a recent systematic review and meta-analysis [54] that included a total of 41 reports (3 RCTs and 38 cohort studies), showing inverse associations with cardiovascular mortality, coronary heart disease incidence and mortality, stroke incidence and mortality and myocardial infarction incidence when comparing the highest versus the lowest categories of MD adherence. However, it only included adult individuals with diabetes, which poses a challenge to generalize to a general or a geriatric population.

We did not find an association between adherence to MD and cardiovascular events in the population >70 years of age. A possible explanation is that all-cause mortality is a more reliable endpoint, making it less subject to bias. The absence of statistical significance may also be partially explained by the limited number of studies available in this age group, which led to increased variability and wider confidence intervals while the point estimate remained substantially unmodified. Moreover, for cause-specific death, the accuracy of recording depends on the correct identification of the cause of death which may be more difficult to identify in older patients, due to multiple comorbidities.

### Strengths and Limitations

The present study has the following strengths: To the best of our knowledge, no other meta-analysis has been undertaken with the aim to investigate the relationship between MD adherence and all-cause mortality and cardiovascular events in a geriatric population, expanding our current knowledge to a demographic beyond the age of 70 through a targeted sub-analysis; and it included recent articles which were not included in previous meta-analyses, reporting a total 714,298 participants from different countries and continents.

However, some limitations should be noted. First, the majority of the studies included had an observational design which might to some extent have affected the quality of the results and contributed to the high heterogeneity of the results. However, we were able to perform sensitivity analysis on the overall cardiovascular events and cardiovascular non-fatal events on RCTs which confirmed our main results. Secondly, it should be noted that the use of different definitions of the MD in terms of different categorizations of the scores, inclusion of different foods, and/or with different cut points may result in misclassification of the degree of adherence to the MD score. Furthermore, we only searched for articles in the Pubmed database; however, it included the largest body of international evidence; thus, the risk of not including international peer reviewed studies is minimal. Finally, a publication bias, which challenges the validity of any meta-analysis, should be taken into account.

## 5. Conclusions

Our findings show that the positive effect of adherence to the MD on all-cause mortality and cardiovascular fatal and non-fatal events observed in the general population is also present in people beyond 60 years of age.

Data on adherence to the MD and cardiovascular outcomes in the oldest old remain inconclusive. Thus, new, large-scale RCTs should be carried out specifically on older people in order to better understand its role in cardiovascular prevention.

## Figures and Tables

**Figure 1 nutrients-16-03947-f001:**
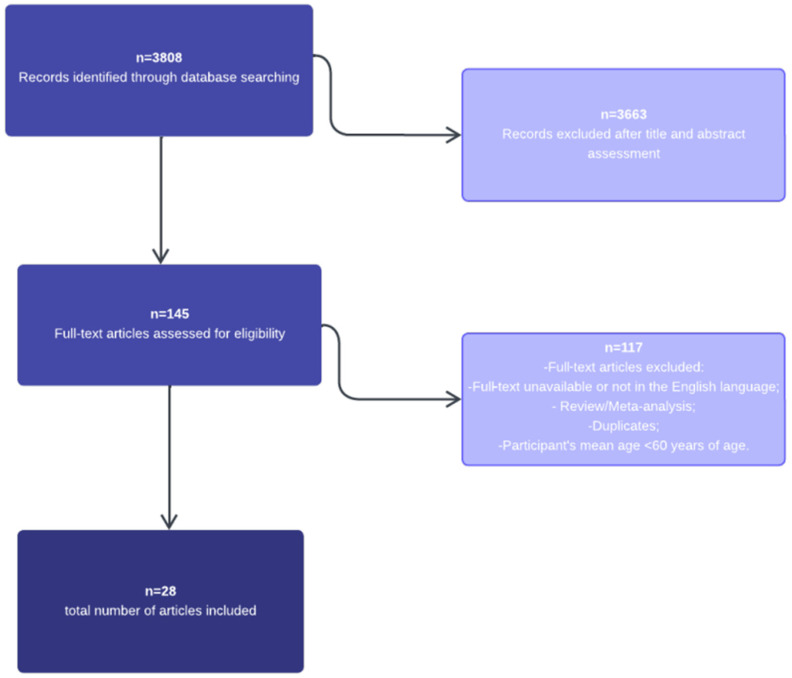
Flow diagram of study selection.

**Figure 2 nutrients-16-03947-f002:**
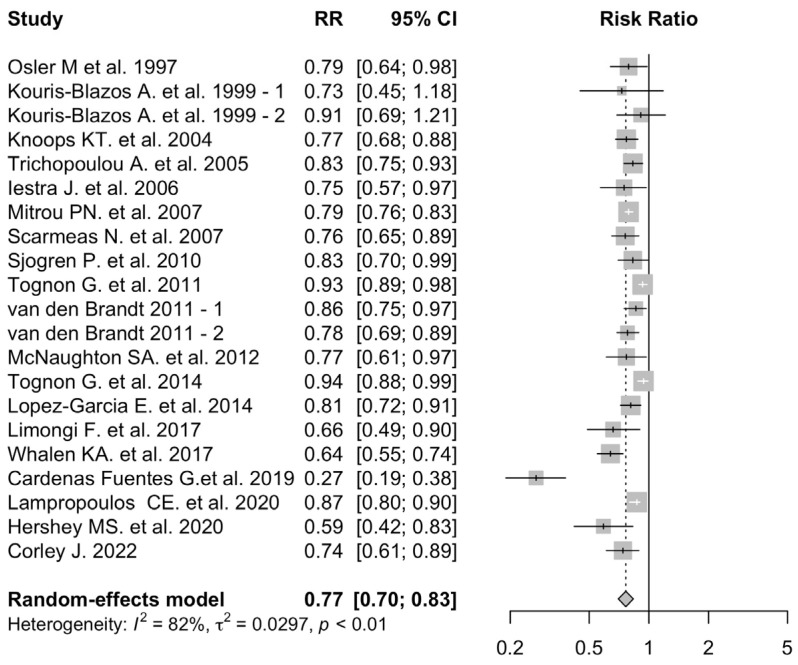
MD adherence and all-cause mortality. Adherence to MD is associated with a reduced all-cause mortality risk [9,28,29,30,31,32,33,34,35,36,37,38,39,40,41,42,43,44,51]. -1 and -2 indicate two arms of the same study.

**Figure 3 nutrients-16-03947-f003:**
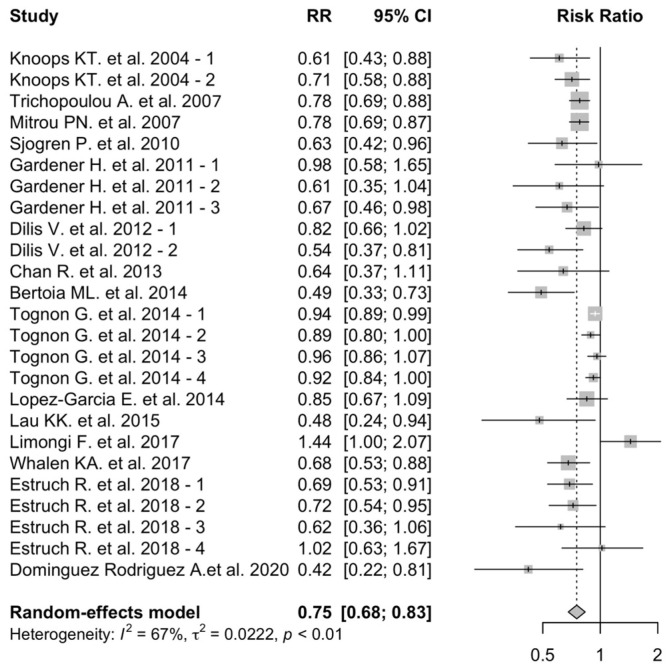
MD adherence and overall cardiovascular events. MD adherence is associated with a reduced risk in overall cardiovascular events [9,23,29,30,37,41,43,45,46,47,48,49,50,51,52]. -1,-2, -3, and -4 indicate different arms of the same study.

**Figure 4 nutrients-16-03947-f004:**
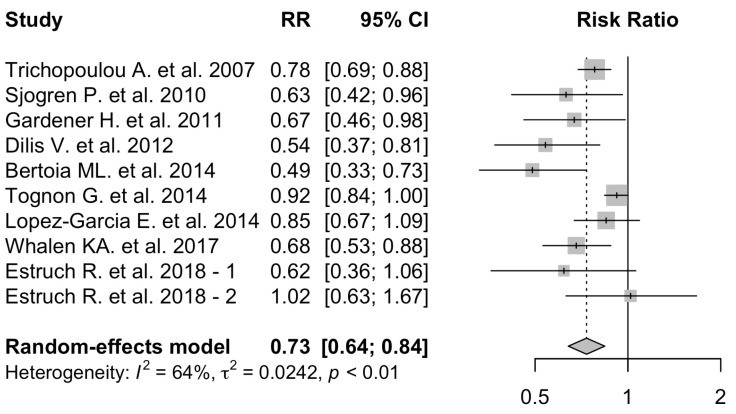
Adherence to MD and cardiovascular mortality. MD adherence is associated with a reduced risk of cardiovascular mortality [23,29,36,37,41,43,47,48,49]. -1 and -2 indicate two arms of the same study.

**Figure 5 nutrients-16-03947-f005:**
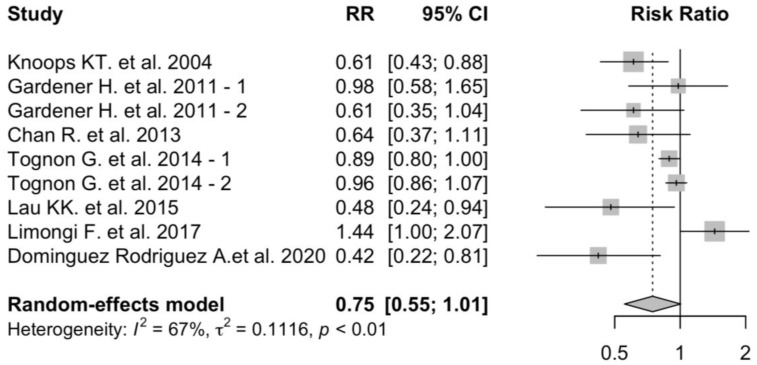
MD adherence and cardiovascular non-fatal events. MD adherence is associated with a reduced risk of cardiovascular non-fatal events [9,30,37,45,48,50,52]. -1 and -2 indicate two arms of the same study.

**Table 1 nutrients-16-03947-t001:** Characteristics of included studies.

Author, Nation, Year	Study Design (Acronym)	Participants (Males)	Mean Age (SD)	Follow Up, Months	MD Adherence	Setting	Outcomes
Cardenas Fuentes G. et al., Spain, 2019 [28]	Cohort (PREDIMED)	7447 (3202)	67 (6.2)	56 (median)	14-Point Mediterranean Diet Adherence Screener (MedDiet score)	Community dwelling	All-cause mortality
Limongi F. et al., Italy, 2017 [9]	Cohort (ILSA)	4222 (1816)	73 (5.4)	85	A priori score based on the Mediterranean pyramid components	Community dwelling	All-cause mortality/ Overall cardiovascular events
Sjogren P. et al., Sweden, 2010 [29]	Cohort (Uppsala Longitudinal Study of Adult Men cohort)	924 (924)	71 (1)	122 (median)	HDI score + MDS + CR score	Community dwelling	All-cause mortality/ Overall cardiovascular events
Knoops KT. et al., Europe, 2004 [30]	Cohort (HALE project)	2339 (1507)	80	156	Dietary History Method	Community dwelling	All-cause mortality/ Overall cardiovascular events (CHD/CVD *)
Iestra J. et al., Europe, 2006 [31]	Cohort (N/A)	426 (284)	75 (2)	120	Modified Mediterranean Diet Score (MDS)	Community dwelling	All-cause mortality
Lampropoulos CE. et al., Greece, 2020 [32]	Cohort (N/A)	183 (91)	79	24	MDS	Hospitalized patients	All-cause mortality
Corley J., Scotland (UK), 2022 [33]	Cohort (The Lothian Birth Cohort 1936)	882 (425)	69.5 (0.8)	144	FFQ (168 item) + MIND score	Community dwelling	All-cause mortality
Hershey MS. et al., Spain, 2020 [34]	Cohort (The Seguimiento Universidad de Navarra cohort)	20,494 (7993)	66 (16.3)	145 (median)	Mediterranean lifestyle index (MEDLIFE, 28 items)	Community dwelling	All-cause mortality
Tognon G. et al., Sweden, 2011 [35]	Cohort (Gerontological and Geriatric Population Studies in Gothenburg)	1037 (497)	70	101	Modified Mediterranean diet score (refined mMDS) + HALE mMDS	Community dwelling	All-cause mortality
Trichopoulou A. et al., Europe, 2005 [36]	Cohort (EPIC study, no CAD history)	74,607 (24,545)	63	88 (median)	FFQ + recall of dietary intake over 24 h	Community dwelling	All-cause mortality
Tognon G. et al., Denmark, 2014 [37]	Cohort (Danish MONICA)	1849 (901)	60	168	MDS	Community dwelling	All-cause mortality/ Overall cardiovascular events/Fatal events
Van Den Brandt, Netherlands, 2011 [38]	Cohort (N/A)	3576 (1690)	62	120	FFQ + aMED	Community dwelling	All-cause mortality
Kouris-Blazos A. et al., Australia, 1999 [39]	Cohort (N/A)	330 (164)	73	12	FFQ (250 items)	Community dwelling	All-cause mortality
Scarmeas N. et al., USA, 2007 [40]	Cohort (N/A)	192 (42)	82.9 (7.7)	52	sFFQ	Community dwelling	All-cause mortality
Lopez Garcia E. et al., USA, 2014 [41]	Cohort (Health Professionals Follow-Up Study and the Nurses’ Health Study)	17,415 (6317)	69 (9)	91 (median)	FFQ	Community dwelling	All-cause mortality/ Overall cardiovascular events
McNaughton SA. et al., United Kingdom, 2012 [42]	Cohort (The British Diet and Nutrition Survey)	972 (515)	74	168	Healthy Diet Score (HDS) + Recommended Food Score (RFS) +Mediterranean Diet Score (MDS)	Community dwelling	All-cause mortality
Whalen KA. et al., USA, 2017 [43]	Cohort (REGARDS study)	21,423 (11,558)	63	75 (median)	FFQ	Community dwelling	All-cause mortality/ Overall cardiovascular events/ Fatal events
Osler M. et al., Denmark, 1997 [44]	Cohort (N/A)	202 (101)	72.2	72	3–day estimated record + FFQ	Community dwelling	All-cause mortality
Papadaki A. et al., Spain, 2017 [22]	RCT (PREDIMED)	7403 (4240)	67 (6.2)	56	14-item Med Diet questionnaire + FFQ	Community dwelling	Overall cardiovascular events/ No Fatal events (HF)
Dominguez Rodriguez A. et al., Spain, 2020 [45]	Cohort (N/A)	284 (204)	73 (3)	12	PREDIMED questionnaire (14-item)	Comminity dwelling	Overall cardiovascular events/ No Fatal events (HF)
Trichopoulou A. et al., Europe, 2007 [46]	Cohort (EPIC, CAD history)	2671 (1833)	65	79	FFQ + 7–14day record on meals consume	Community dwelling	Overall Cardiovascular events/Fatal events
Estruch R. et al., Spain, 2018 [23]	RCT (PREDIMED)	7447 (3202)	67 (6.2)	56 (median)	14-item Med Diet questionnaire + FFQ	Community dwelling	Overall Cardiovascular events/ Fatal events
Bertoia ML. et al., USA, 2014 [47]	Cohort (Women’s Health Initiative study)	93,122 (0)	65 (6.4)	125	FFQ	Community dwelling	Overall Cardiovascular events/ Fatal events
Gardener H. et al., USA, 2011 [48]	Cohort (The Northern Manhattan Study)	2568 (924)	69 (10)	108	Modified Block National Cancer Institute food-frequency questionnaire	Community dwelling	Overall Cardiovascular events/Fatal/No Fatal events
Dilis V. et al., Greece, 2012 [49]	Cohort (EPIC study, no CAD history)	23,939 (9740)	60	120	FFQ (200 item)	Community dwelling	Overall Cardiovascular events/Fatal events
Lau KK. et al., Hong Kong, 2015 [50]	Cohort (N/A)	274 (212)	68 (10)	7.7	MDS	Community dwelling	Overall Cardiovascular events/No Fatal events
Mitrou PN. et al., USA, 2007 [51]	Cohort ((NIH)-AARP Diet and Health Study)	380,296 (214,284)	62	120	FFQ	Community dwelling	Overall Cardiovascular events
Chan R. et al., Hong Kong, 2013 [52]	Cohort (N/A)	2735 (1338)	72.2(5.4)	68	FFQ	Community dwelling	Overall Cardiovascular events/ No Fatal events (CVD */Stroke)

FFQ: food frequency questionnaire; MDS: mediterranean diet score; RCT: randomized controlled trial; HDI: healthy diet indicator; CHD: coronary heart disease; CVD: cardiovascular disease; HF: heart failure; * as defined by International Classification of Diseases, ninth revision (ICD-9) coding system. Follow up is expressed in mean months, unless specified otherwise.

## Data Availability

The Dataset is available on request from the authors. The data are not publicly available due to privacy.

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
