# Peer review of "Mediterranean Diet in Older Adults: Cardiovascular Outcomes and Mortality from Observational and Interventional Studies—A Systematic Review and Meta-Analysis"

_nutrients, 2024, doi:10.3390/nu16223947_

Round 1
Reviewer 1 Report
Comments and Suggestions for Authors
The authors provide an SRMA on the effects of the Mediterranean diet on CVD outcomes and mortality.
The overall rationale is clear from the paper, but not the title. It needs to specify, if observational or interventional studies are used.
Introduction: Fine.
Methods: Line 81 ff.: If RCTs contain multiple treatment arms, every included active treatment arms needs to be compared to the control group with its patient number divided by the number of included active treatment arms (i.e. in the case of PrediMed, both treatment arms are used with full "n", but the control group gets "n/2" per comparison).
Why was there no study protocol?
Why was there no prior registration?
Line 90: How many studies were left out due to unobtainable standard error?
Table 1: The PrediMed study was included five times as individual RCT. This is apparently inacceptable.
Table 1: The PrediMed study was included another two times as cohort study. Inacceptable, once again.
Table 1: Please clarify, that there a no duplicate studies among the observational studies. Confirm this by noting the study name in table 1.
Further analysis: RCTs and observational studies were pooled together. An error like this rules out publication at all. The authors should consult experts on SRMAs before another attempt of publication.
Discussion: Based on massive flaws in study methodology.
Comments on the Quality of English Languageminor changes needed
Author Response
The authors provide an SRMA on the effects of the Mediterranean diet on CVD outcomes and mortality.
The overall rationale is clear from the paper, but not the title. It needs to specify, if observational or interventional studies are used.
The title has been revised to better reflect that the study encompasses both observational and interventional studies, providing greater clarity on the scope of the work.
Introduction: Fine.
Methods: Line 81 ff.: If RCTs contain multiple treatment arms, every included active treatment arms needs to be compared to the control group with its patient number divided by the number of included active treatment arms (i.e. in the case of PrediMed, both treatment arms are used with full "n", but the control group gets "n/2" per comparison).
We apologize for an incomplete reporting of the methods used to analyse our data. We used a three-level approach as suggested by Cheung (“Modeling Dependent Effect Sizes with Three-Level Meta-Analyses: A Structural Equation Modeling Approach.” Psychological Methods 19 (2): 211, 2014.) that takes into account the nesting of comparisons within the same study. We specified this in the Methods section (lines 99-101).
Why was there no study protocol?
Why was there no prior registration?
We recognize that the study protocol should be registered: it is now registered in INPLASY (International Platform of Registered Systematic Review and Meta-analysis Protocols) with registration number INPLASY2024100072 and DOI number 10.37766/inplasy2024.10.0072 (see Methods, lines 95-96).
Line 90: How many studies were left out due to unobtainable standard error?
We did not exclude studies for this reason. We have now specified it in the Results (lines 128-129).
Table 1: The PrediMed study was included five times as individual RCT. This is apparently inacceptable.
The inclusion of different studies with same subjects but with different outcomes was based on paper by Trikalinos et al that shows that a univariate approach to this situation yields basically similar results compared to a multivariate analysis (Statist. Med. 2014, 33 1441–1459). However, we acknowledge that not all the studies we included satisfied the required assumptions, therefore we decided to modify analysis accordingly.
Table 1: The PrediMed study was included another two times as cohort study. Inacceptable, once again.
Please see the reply to the previous comment.
Table 1: Please clarify, that there a no duplicate studies among the observational studies. Confirm this by noting the study name in table 1.
We added the study name to table 1 where available.
Further analysis: RCTs and observational studies were pooled together. An error like this rules out publication at all. The authors should consult experts on SRMAs before another attempt of publication.
The pooling of RCT and observational studies is admittedly a limitation of this study, but it is not an error and is indeed considered in the Cochrane Handbook for Systematic Reviews (see chapter 24). When observational studies are expected to provide information on real-life effect of data, the pooling may be advantageous, and the bias introduced may be acceptable.
Discussion: Based on massive flaws in study methodology.
We are confident that our explanation of the methodological choices and the revision done will lead to a different evaluation.
Reviewer 2 Report
Comments and Suggestions for Authors
Dear author,
Thank you to give the opportunity to review your manuscript. I have some suggestion to improve it.
Introduction.
In general, is weak more studies relating elderly, cardiovascular diseases and Mediterranean Diet should be included.
References 5 is general, should be change for other more related to Mediterranean Diet.
Material and methods
Line 64 should be defined the start date of the search.
Should be amplify the concept (MOOSE) that is introduced in the abstract.
Results
In the tables (Author, Nation, Year) the references of the studies should be included also in the references. Example (Cardenas Fuertes G. et al., Spain, 2018 [21]).
Line 151, n.5 should be between brackets (n.5). Apply in the rest.
Discussion
Should be rewrite including the references of the studies evaluated in the table 1.
References
Low number of references for a systematic review and Meta-Analysis.
Best regards,
Author Response
Introduction.
In general, is weak more studies relating elderly, cardiovascular diseases and Mediterranean Diet should be included.
We thank the reviewer for the suggestion: we included other studies references in the Introduction.
References 5 is general, should be change for other more related to Mediterranean Diet.
We changed the reference as suggested.
Material and methods
Line 64 should be defined the start date of the search.
We specified the start date of data search (line 66).
Should be amplify the concept (MOOSE) that is introduced in the abstract.
We reported it in the Methods (lines 89-91).
Results
In the tables (Author, Nation, Year) the references of the studies should be included also in the references. Example (Cardenas Fuertes G. et al., Spain, 2018 [21]).
We reported the references of the studies.
Line 151, n.5 should be between brackets (n.5). Apply in the rest.
We modified the text accordingly to the reviewer’s comment.
Discussion
Should be rewrite including the references of the studies evaluated in the table 1.
We modified the references accordingly.
References
Low number of references for a systematic review and Meta-Analysis.
We agree with the reviewer. We included a higher number of references in the revised manuscript. Therefore, we believe that now the manuscript has an adequate number of references.
Round 2
Reviewer 1 Report
Comments and Suggestions for Authors
The authors have responded to the reviewer's remarks.
However, the major methodological flaw by combining observational studies and RCTs in one pooled SRMA is still remaining. Such an analysis is not acceptable.
minor
Reviewer 2 Report
Comments and Suggestions for Authors
Dear Authors,
Changes have been applied correctly.
Best regards,